# Mechanical Behavior of Porous Functionally Graded Nanocomposite Materials

**A. F. Mota [1] and M. A. R. Loja [1,2,\*]** 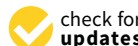

[1] CIMOSM, ISEL, IPL—Centro de Investigação em Modelação e Optimização de Sistemas Multifuncionais, Instituto Politécnico de Lisboa, 1959-007 Lisboa, Portugal; amota@dem.isel.pt

[2] IDMEC—Instituto Superior Técnico, Universidade de Lisboa, 1049-001 Lisboa, Portugal

\* Correspondence: amelialoja@dem.isel.ipl.pt; Tel.: +351-218-317-000

**Abstract:** Materials used in biomedical applications need to cope with a wide set of requisites, one of them being their structural adequacy to a specific application. Thus, it is important to understand their behavior under specified standard cases, namely concerning their structural performance. This objective constituted the focus of the present study, where nanocomposite functionally graded materials integrating different porosity distributions were analyzed. To this purpose a set of numerical simulations based on the finite element method, reproducing American Society for Testing and Materials (ASTM) tensile and bending tests were considered. The results obtained show a good performance of the models implemented through their preliminary verification. It is also possible to conclude that carbon nanotubes and porosity distributions provide different and opposite effects in the context of the nanocomposite materials analyzed.

**Keywords:** functionally graded materials; carbon nanotubes; porosities; numerical simulation of ASTM tests

## 1. Introduction

The understanding and manipulation of materials over time have been fundamental for technological development, raising the importance of their innovation for economic and environmental reasons, among others. According to A. Sola et al. [1], materials with a functional gradient are an excellent example of innovation in the materials sector as they are designed to satisfy predefined objectives. The particular characteristic that allows the projection of these functionally graded materials (FGMs) in function of the objectives traced is the variation of their composition and/or their microstructure in one or more spatial directions according to a given law [1,2].

In China, in the late 1980s, the first FGM was formulated in the development of new coatings for thermal insulation [2]. Another reason for the development of this type of material was the need to minimize the stress discontinuities verified in the laminated composite materials [3]. These versatile materials have several advantages including reduced in-plane and transverse stresses, low residual stresses, high thermal resistance, low thermal conductivity, and high fracture toughness. Additionally and contrarily to classical laminated composites, FGMs do not exhibit severe interlaminar stresses because of the smooth variation of material composition [4].

Another function of relevant importance found through the application of functionally graded materials as coatings is the reduction of mechanical vibrations. Catania and Strozzi studied a multi-layer beam model with a metal core coated with thin layers of variable stiffness, considering the existence of a local dissipation mechanism at the layers' interfaces. From this study they concluded that the damping of a uniform beam of rectangular section, vibrating in flexural-axial plane conditions, can be related to the application of a superficial coating. More particularly, the application of these coatings

may cause increased damping of thin-walled components under flexural vibration conditions. In spite of that, experimentally it was found that most of these coating surface treatments have a negligible effect in the vibrational damped response when the components are subjected to testing or operating conditions [5].

Also in the context of damping, Yu et al. [6] reveals that in a coated mechanical system there is an optimum coating thickness which leads to an optimum ratio between stiffness and damping. In particular, increasing the difference between the Young's modulus of the substrate and the coating allows for better damping performance and rigidity of the system. The reduction of vibrations is advantageous in several mechanical systems, since it allows an increase in the service life and a reduction in its weight. In a previous work [7], Loja studied, among others, a sandwich beam with graded metal– graphene nanocomposites skins and a soft core, since soft and lightweight cores allow a viscoelastic damped behavior. This kind of behavior is of great interest in most engineering applications not only because it is responsible for the vibration attenuation but also for greater fatigue and impact resistance.

In fact, FGMs are not new in nature. In terms of flora and fauna, we find several examples such as bamboo or even the shells of mollusks with a characteristic hierarchical structure. If we now think of the human body, more precisely the constitution of the skin, its three layers—epidermis, dermis and hypodermis—constitute a multilayer system. This system is also an FGM by itself, having several functions such as protecting the body from mechanical impacts and pressure, variations in temperature, micro-organisms, radiation, and chemicals. Still with respect to the human body we have the tooth, whose surface is constituted by hard enamel with crystals of hydroxyapatite being its inner dentin, composed of fibers of collagen and hydroxyapatite. Its constitution and the gradual transition between the inner and the superficial material make the tooth another example of FGM in the human body. Being a type of material so common in nature and in particular in the human body, the FGMs have been extensively developed for most diverse biomedical applications [2].

Most of the man-made FGMs have a metal or ceramic base. However, polymer-based composites may lead to better mechanical and physical properties in a wide range of applications. Tissue engineering has shown great interest in polymeric materials, particularly those of porous structure because they have properties that evolve gradually. As stated by Naebe and Shirvanimoghaddam, the microstructure with gradual evolution in nano-reinforced polymer composites has the capacity to expand the engineering applications of these materials through the development of new composites with customized properties. In particular, the inclusion of nanomaterials like carbon nanotubes (CNTs) can lead to multi-functional materials with better mechanical, physical, and chemical performances for biological applications [1].

Due to the capacity to replicate different tissues, porous materials have been recently investigated in different areas of science and engineering. Bone scaffolds are just one interesting example that requires simultaneously high porosity levels and connectivity, which are essential characteristics for healthy bone growth. Some literature studies [2,8] mentioned the importance of porosity fraction and pore dimensions on scaffolds for bone formation, as well as the influence that shape, distribution, and porosity level has on mechanical properties. A graded porous biomaterial, made of polyurethane and Bioglass ® 45S5 microfiber, is usually used in tissue engineering [2]. In this case, the polyurethane is responsible for the elasto-plastic behavior, while the bioglass reinforcement guarantees mechanical strength and stiffness as well as good bone adhesion.

Multiscale hierarchic materials are becoming an effective and advantageous option due to the scientific and manufacturing technological advances. Carbon nanotubes have captured the interest of the scientific and technical society due to their singular properties, particularly the physical and mechanical ones, so CNT reinforced polymer composites have been researched and developed [9]. Biomedical applications, such as tissue regeneration and diagnostic techniques, have taken advantage of the characteristics presented by CNTs. However, their applicability in biomedicine still requires that the long-term non-toxicity and biodegradability of CNTs is ensured [10].

Being the strongest and stiffest materials today, single-walled carbon nanotubes (SWCNTs) and multi-walled carbon nanotubes (MWCNTs) result from graphene single or multiple sheets convolved into cylinders [11]. The morphological diversity of SWCNTs includes length, diameter, chirality, and configuration. Since MWCNTs can be considered as a conjugation of SWCNTs, their morphological variability is greater. The remarkable resilience of SWCNTs is clearly superior to that of metals and carbon fibers when subjected to the same load. Together with other exceptional properties, it makes carbon nanotubes an excellent choice as reinforcement of composites in the most diverse structural and functional applications. CNTs were first used as reinforcement of a polymer nanocomposite in 1994 and since then have been the preferred reinforcement in several polymer matrix composites [12]. Despite the impressive advantages of the CNTs already mentioned, their properties can still be improved through developments in dispersion and reinforcement-matrix adhesion [9].

According to the literature [9], the path in search of the perfect polymer matrix nanocomposite led to the study of its mechanical properties, since these were below the expectations dictated by the law of mixtures. The concentration of CNTs required for them to act as reinforcement in the polymer matrix and to allow the usufruct of the matrix properties is very low. The loading of CNTs has a strong influence on the mechanical reinforcement response, which is limited to high loads due to the high viscosity of the polymer matrix and the void defects. For this reason, it is still necessary to deepen the research behind a solution for this, although the functionalization of CNTs evidences improvements in the dispersion level and, consequently, in the mechanical properties of these nanocomposites [9].

In a complementary domain, it is important to refer the studies developed at the nanoparticles level, which are relevant not only from a multiscale perspective, but also from the point of view of these structures by themselves as at nanoscale it is necessary to consider forces that are negligible to the macroscale. For this reason, the valid mechanical principles for macrostructures are not suitable for application to nanostructures. In this sense, new theories have arisen around the characteristic nonlocal nature of these structures [13] The nonlocal theories of continuum mechanics are often used to take into account the size effects present in nanostructures behavior [14]. To most classical continuum theories, stresses at one point depend only on the extensions at that point, but according to the nonlocal theories these stresses are a function of the extensions of all points of the material. In fact, these theories take into account the interatomic forces, introducing an internal length scale as a material parameter into the constitutive equations [15]. The Eringen differential model has been widely applied in the study of nanostructures in order to explain the size dependence that these structures demonstrate. According to Barreta et al. [13], the Eringen strain-driven nonlocal integral theory is not suited to structural mechanics, unlike the recent stress-driven nonlocal integral theory. According to this recent theory, the nonlocal elastic strain field results from the convolution between the stress field and a suitable average kernel. According to the Apuzzo et al. study [14], the stress-driven integral model provides a great improvement in the mechanical behavior study of nanostructures since all boundary conditions are provided by the theory itself, allowing a way to overcome the constraints of Eringen's theory with these kinds of structures. In addition to stress-driven nonlocal integral theory [13], new models of nonlocal elasticity have recently been developed, of which the integro-differential and the two-phase models are examples [16].

With the present work it was intended to analyze the mechanical behavior of a nanocomposite FGM resulting from the graded mixture of MWCNTs and polysulfone (PSU) which is a polymeric material considering that this adjustable mixture can be affected by different levels and distributions of porosities. The different materials obtained were further analyzed in the light of two ASTM tests of numerical simulations, namely a tensile and a three-point bending test. The results obtained are discussed in order to conclude on the performance of these materials considering the influence of specific parameters related to FGM distribution, to the porosities distribution, and the MWCNTs graded distribution.

## 2. Materials and Methods

### 2.1. Materials Overview in Biomedical Applications

Materials play a very important role in the feasibility of biomedical a~l applications. To illustrate this aspect, one can refer for example to some aspects associated with bone tissue. This tissue degrades over time due to the simple aging effect, with possible diseases accelerating this process. For this reason, the use of implants or prostheses in this field is recurrent, and the success of this practice depends largely on the material properties, which include mechanical behavior. The implemented materials must then be selected in order to allow biofunctional balance, related to bone growth and its integration, as well as a good biomechanical performance with respect to stiffness and elasticity. Regarding the biomechanical performance, porous materials present a smaller Young's modulus when compared to the widely used materials [17] however they are able to ensure a better integration in the human body.

As reported in Section 1, a porous composite material made of Bioglass ® 45S5 microfiber reinforced polyurethane matrix is used for biomedical applications, particularly in tissue engineering [2]. The ceramic of hydroxyapatite, produced by sintering, has a porous structure very similar to that present in the human bone, the reason why it has been widely used in bone implants. As the porous metal surfaces provide a faster and stronger fixation, hydroxyapatite has been used as a coating for these surfaces, and improvements have been made at this level [18].

With the advances in nanotechnology, the applications of nanomaterials have been under extensive development. In this field, CNTs are the most selective candidate for a great variety of multifunctional biomedical applications due to their peculiar properties, e.g., biomedical sensing and therapeutics. In therapeutics applications, CNTs are used as drug and gene delivery for many disease treatments, as well as in vaccine delivery, neuroprosthetic implants, and tissue engineering as well as regenerative medicine like bone, muscle or neural regeneration. Another field of CNTs applications is the diagnosis field, where this interesting material is used for biomedical imaging, and for biomolecular detection. CNTs are also used as biosensors, and as nanotweezers [19]. PEGylated SWCNTs and mesoporous silica-coated SWCNTs are used as photothermal agents, being the first ones also used for bioimaging [20] and only two examples of the enormous variety of composite carbon nanotubes materials applied in biomedicine.

One of the biomedical applications of polymer matrix composites resides in the production of membranes for artificial organs. Guided tissue regeneration (GTR) and, more recently, guided bone regeneration (GBR) are techniques for which these membranes are of particular interest as they act as barriers to regeneration. Polysulfone, being a non-resorbable polymer, presents itself as a good candidate for polymer matrix. This matrix reinforced with CNTs may have a promising future in the use of membranes in biomedical applications, and it has been found that the maximum addition of 1.5% (wt) MWCNTs promotes an increase of the pore size, whereas its presence at 4% (wt) increases rejection and flow [21]. CNTs have also been applied as reinforcement in FGMs, which are also used in biomedical applications, such as prostheses and bone implants [22].

As reported, CNTs have a wide range of applications, particularly in the field of biomedicine. However, their reactivity and toxicity are a reality. These detrimental characteristics result largely from manufacturing processes. Knowing the role that functionalization with biocompatible materials represents in toxicity, polyethylene glycol reinforced with small size MWCNTs composites can be considered a good option in reducing toxicity. Despite the disadvantages, CNTs have been successfully used in various biomedical applications through their functionalization to reduce toxic effects [19]. Continuous research on CNTs will allow a better understanding of the biological, physical, and chemical processes, which can lead to a wider use of these materials in nanobiomedicine. In addition, the biodegradability aspect of CNTs should be studied more deeply [23].

With this brief overview, the need becomes clearer to address specific aspects of the materials that will be used in a biomedical context. From the structural perspective these materials also must be able

to perform accordingly, thus it is important not only to predict their mechanical properties but also to understand how the structures will behave under standard loading situations. Therefore, in the following sub-sections one focus is on particular aspects related to the materials modelling and the analysis method considered in the present work.

*2.2. Functionally Graded Materials*

FGMs became in recent years quite attractive due to their flexibility on tailoring their material properties which make them suitable materials for a great variety of applications in fields like aerospace, nuclear, and bioengineering [1]. Figure 1 illustrates a set of different FGMs application areas among others.

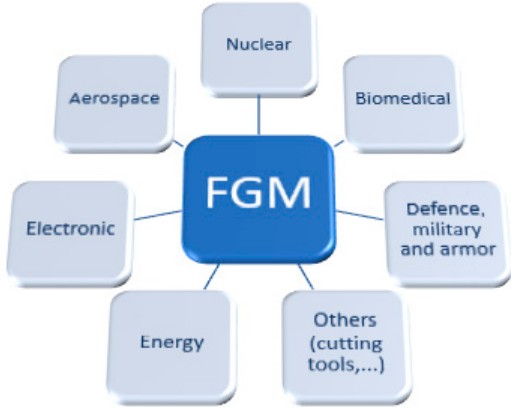

**Figure 1.** Functionally graded materials (FGMs) applications.

FGMs are commonly manufactured to present a predetermined spatial distribution of the phases that constitute them, usually metal, ceramic or polymer phases. This spatial variation of composition occurs in a gradual and continuous manner, resulting in specific morphologies and properties that distinguish them from others [1]. To illustrate this material concept, a dual-phase FGM, with its mixture grading in one spatial direction, is presented in Figure 2. There are several modes of gradual evolution of constitution, and some applications may contain variations in more than one direction [2].

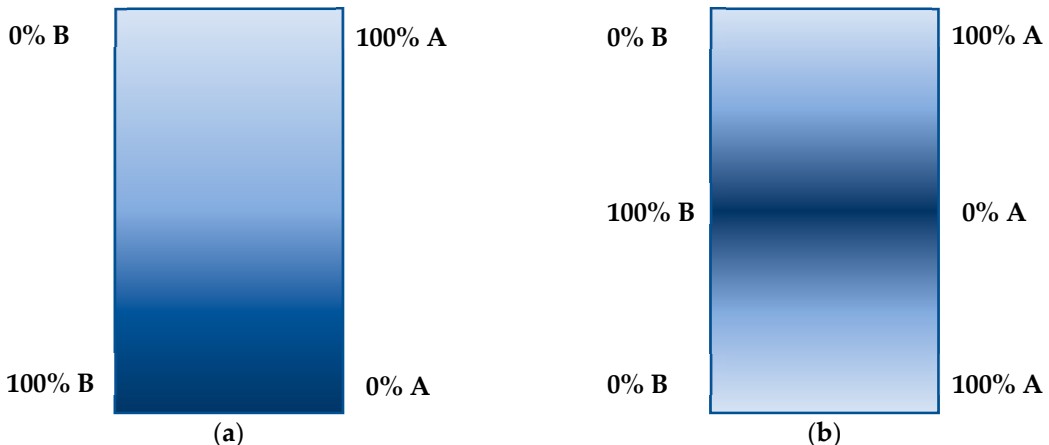

**Figure 2.** Two examples of dual-phase FGM mixture variation along the thickness direction, (**a**) lower surface to upper surface, (**b**) center to outer surfaces.

As the bulk fractions of the constituents of a FGM and its own microstructure evolve gradually, also their macroscopic properties, such as hardness, undergo continuous and gradual changes. In the case of an $Al_2O_3/Y\text{-}ZrO_2$ FGM, it is possible to observe the mutual presence of the properties of both

phases, namely alumina hardness and zirconia strength. For this reason, this FGM has been the subject of several studies regarding biomedical applications, particularly for hip and knee prostheses thanks to the great biocompatibility. The potential to design FGMs with the required properties at mechanical, physical or biological levels, makes these materials excellent candidates for the fabrication of numerous products [1]. As stated, the volumetric fraction of the constituents of the FGMs changes gradually, this change occurring in a single direction or even in more than one direction at the same time. In this work, a dual-phase FGM plate was considered where the volume fraction depends only on the thickness direction. To this purpose, the volume fraction is considered to vary along the *z*-axis according to the power law:

$$V_f = \left( \frac{z}{h} + \frac{1}{2} \right)^p \tag{1}$$

where *h* represents the plate's thickness and *z* stands for the thickness coordinate, whose origin corresponds to the middle surface of the composite FGM plate, so $z \in [-h/2, h/2]$. This case corresponds to the material distribution represented in Figure 2a. The exponent *p* allows volume fraction behavior adjustment throughout the composite thickness. This behavior in order of exponent *p* can be observed in Figure 3.

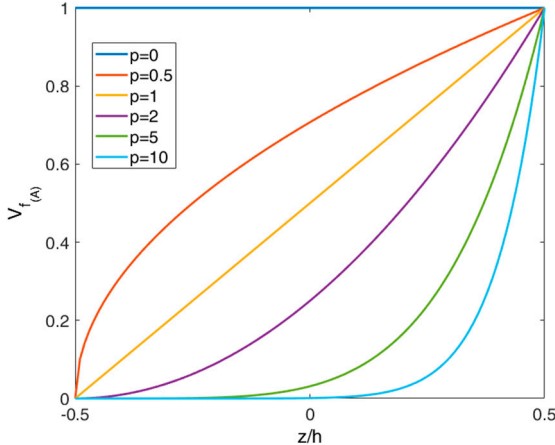

**Figure 3.** Through-the-thickness volume fraction variation.

In this specific case, the volume fraction of the FGMs constituents varies with the z coordinate, whereby the corresponding material properties will also vary continuously along the *z*-direction. The Voigt rule of mixtures allows the distribution of the elastic properties to be determined along the thickness of a FGM plate by the following expression:

$$P_{(FGM)} = P_{(A)} V_{f_{(A)}} + P_{(B)} V_{f_{(B)}} \text{ with } V_{f_{(A)}} + V_{f_{(B)}} = 1 \tag{2}$$

where $P_{(FGM)}$ is the FGM property, and $P_{(A)}$ and $P_{(B)}$ represent the corresponding property of phases A and B, respectively. The relation in (2) allows a prediction for the effective properties of a graded material to be obtained. This approach is appropriate in cases where the contrast between phases is not very marked [24].

*2.3. Porosities Distribution*

Functionally graded porous materials are composites that combine the advantages of FGMs with the characteristics of porous materials. These materials have remarkable characteristics, of which the rigidity-weight ratio stands out. The ability to provide unique mechanical properties with a relatively light weight makes porous FGM a widely implemented material in a variety of engineering areas, including the mechanics and biomedical areas. Due to their possible application in diverse systems,

this particular material has captured the interest of research groups with regard to development and implementation in real life engineering [3,25].

The manufacturing methods of FGMs are a developing area, the sintering process being the most used due to its cost-benefit ratio. However, the sintering process enables the formation of micro-voids or porosities. Despite the significant improvement observed recently in this area, porosity remains a frequent defect in FGMs [3].

As related above, porosities are a consequence of the manufacturing process and reduce the material's strength. Hence, the effects of this kind of defect need to be considered when studying the mechanical behavior of FGMs structures. With this concern, N. V. Nguyen et al. [26] worked on the mechanical response of porous FGM plates, considering for this purpose two distinct porosity distributions along the thickness direction, namely even and uneven. Figure 4 illustrates these different porosity distributions in an FGM plate like the one represented in Figure 2a.

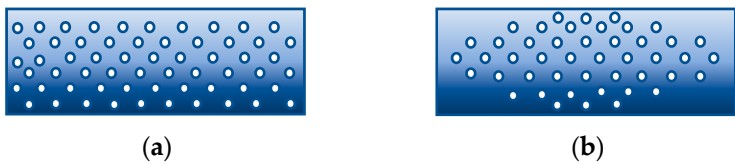

(a)          (b)

**Figure 4.** Porosity distributions: (**a**) even; (**b**) uneven.

Similar to the rule of mixtures (2), the effective material properties evolution throughout the thickness including the effect of an even distribution of porosities is given by:

$$P_{(FGM)}(z) = \left(P_{(A)} - P_{(B)}\right)\left(\frac{1}{2} + \frac{z}{h}\right)^{p} + P_{(B)} - \frac{\alpha}{2}\left(P_{(A)} + P_{(B)}\right) \tag{3}$$

When considering porosities distributed unevenly along the plate thickness direction, the effective properties can be determined as follows:

$$P_{(FGM)}(z) = \left(P_{(A)} - P_{(B)}\right)\left(\frac{1}{2} + \frac{z}{h}\right)^{p} + P_{(B)} - \frac{\alpha}{2}\left(P_{(A)} + P_{(B)}\right)\left(1 - \frac{2|z|}{h}\right) \tag{4}$$

In both relations—(3) and (4)—$\alpha$ is a porosity volume fraction, so $0 \leq \alpha \leq 1$. For $\alpha = 0$, the FGM plate has no porosities, which is usually called the perfect FGM plate [26]. Also, J. Kim et al. [25] referred that, being fundamental elements in structural systems and others, the mechanical responses of beams, plates, and shells should be under investigation with regard to FGMs with distributed porosities.

Porous structures have been studied by some researchers, such as Kiran and Kattimani [3], who worked on the influence of porosity distributions on the behavior of structural plates. Their studies revealed a significant effect of porosity on natural frequencies, displacements, and tensions. In a recent work, S. Merdaci and H. Belghoul [27] stated the importance of taking into consideration the porosity effect when designing functionally graded components. To this purpose, they worked on the bending behavior of porous functionally graded plates, aluminum being the metal phase, and alumina the ceramic phase, concluding that the inclusion of porosity parameter increases the deflection and the transverse shear stresses, and decreases the axial stress. However further studies need to be considered namely in the specific context of materials used for biomedical applications.

*2.4. Carbon Nanotubes*

Continuing the above, a multiscale polymer composite reinforced with MWCNT was taken into account in this work. Thus, it was required to first address the prediction of the materials properties involved. The tensile modulus of this nanocomposite can be estimated by using the Halpin Tsai equation, adequate for the nano-inclusions characteristics (5a) to (5c) [28]:

$$E_{(NC)} = \frac{E_{(PM)}}{8}\left[5\left(\frac{1 + 2\beta_{dd}V_{f_{(CNT)}}}{1 - \beta_{dd}V_{f_{(CNT)}}}\right) + 3\left(\frac{1 + 2\left(\frac{l_{(CNT)}}{d_{(CNT)}}\right)\beta_{dl}V_{f_{(CNT)}}}{1 - \beta_{dl}V_{f_{(CNT)}}}\right)\right] \tag{5a}$$

$$\beta_{dl} = \frac{\frac{E_{(CNT)}}{E_{(PM)}} - \frac{d_{(CNT)}}{4t_{(CNT)}}}{\frac{E_{(CNT)}}{E_{(PM)}} + \frac{l_{(CNT)}}{2t_{(CNT)}}} \tag{5b}$$

$$\beta_{dd} = \frac{\frac{E_{(CNT)}}{E_{(PM)}} - \frac{d_{(CNT)}}{4t_{(CNT)}}}{\frac{E_{(CNT)}}{E_{(PM)}} + \frac{d_{(CNT)}}{2t_{(CNT)}}} \tag{5c}$$

where $E_{(NC)}$ are the nanocomposite tensile modulus, and $E_{(CNT)}$ and $E_{(PM)}$ indicates the Young's modulus of the CNTs and isotropic polymeric matrix, respectively. $V_{f(CNT)}$ and $V_{f(PM)}$ represent the volume fractions of CNTs and polymeric matrix, respectively. The geometric characteristics of CNTs are indicated by $l_{(CNT)}$, $d_{(CNT)}$ and $t_{(CNT)}$, representing the length, the outer diameter, and the thickness of CNTs, respectively. The volume fraction ofr the CNTs is given by Equation (6) [28]:

$$V_{f_{(CNT)}} = \frac{w_{(CNT)}}{w_{(CNT)} + \frac{\rho_{(CNT)}}{\rho_{(PM)}}\left(1 - w_{(CNT)}\right)} \tag{6}$$

where $w_{(CNT)}$ represents the weight fraction of the CNTs, and $\rho_{(CNT)}$ and $\rho_{(PM)}$ indicate the mass densities of the CNTs and polymeric matrix, respectively. The Voigt rule of mixtures allows the calculation of the Poisson's ratio of the nanocomposite herein considered, using Equation (7) [29], being $v_{(NC)}$, $v_{(CNT)}$, and $v_{(PM)}$, the nanocomposite, the CNT, and the polymeric matrix Poisson's ratios, respectively.

$$v_{(NC)} = v_{(CNT)}V_{f_{(CNT)}} + v_{(PM)}\left(1 - V_{f_{(CNT)}}\right) \tag{7}$$

Thus, knowing the geometric properties of MWCNTs and the materials properties of the isotropic polymeric matrix and respective reinforcement, it is then possible to estimate the Young's modulus of the polymeric nanocomposites. This prediction is behind the assumption of a good dispersion of CNTs, which is imperative to ensure a good mechanical response of a composite material. This assumption, implicit in the isotropic hypothesis for the nanocomposite studied in this work, can be reached considering low weight/volume fractions of the CNTs [29].

*2.5. Finite Element Model*

The behavior of these functionally graded nanocomposites with porosities was assessed via a set of numerical simulations reproducing the specimen characteristics used on two ASTM tests, namely in a tensile test (ASTM D3039) and in a three-point bending test (ASTM D 7264). These simulations were performed through finite element analyses based on the first order shear deformation theory (FSDT), whose code was also developed and implemented via the use of the bi-linear quadrilateral element [30]. The displacement field corresponding to the FSDT for the analysis of plate type structures, as is considered in the simulations, can be written as:

$$\begin{aligned}
u(x,y,z) &= u^0(x,y) + z.\theta_x^0(x,y) \\
v(x,y,z) &= v^0(x,y) + z.\theta_y^0(x,y) \\
w(x,y,z) &= w^0(x,y)
\end{aligned} \tag{8}$$

where $u(x,y,z)$, $v(x,y,z)$ and $w(x,y,z)$ denote the displacements of a plate point along the coordinates system, being $u^0(x,y)$, $v^0(x,y)$ and $w^0(x,y)$ the corresponding displacements of a mid-plane point. $\theta_x^0$

and $\theta_y^0$ represent the normal to the mid-plane rotations about the y and x axes, respectively. As it can be seen, this model includes five degrees of freedom per node, $q = \left[u^0, v^0, w^0, \theta_x^0, \theta_y^0\right]$.

Omitting dependencies for simplicity questions, the strain field can be written from the Elasticity kinematic relation for small deformations as:

$$
\begin{bmatrix} \varepsilon_x \\ \varepsilon_y \\ \gamma_{xy} \\ \gamma_{yz} \\ \gamma_{xz} \end{bmatrix} = \begin{bmatrix} \frac{\partial u^0}{\partial x} \\ \frac{\partial v^0}{\partial y} \\ \frac{\partial u^0}{\partial y} + \frac{\partial v^0}{\partial x} \\ \theta_y^0 + \frac{\partial w^0}{\partial y} \\ \theta_x^0 + \frac{\partial w^0}{\partial x} \end{bmatrix} + z \begin{bmatrix} \frac{\partial \theta_x^0}{\partial x} \\ \frac{\partial \theta_y^0}{\partial y} \\ \frac{\partial \theta_x^0}{\partial y} + \frac{\partial \theta_y^0}{\partial x} \\ 0 \\ 0 \end{bmatrix} \tag{9}
$$

The in-plane normal strains and the shear strain are represented by $\varepsilon_x$, $\varepsilon_y$, and $\gamma_{xy}$, respectively, the others being denoted by $\gamma_{yz}$ and $\gamma_{xz}$, the interlaminar transverse shear strains. This theory is characterized by the thickness inextensibility, expressed by $\varepsilon_z = 0$. It is important to note that although the materials analyzed are heterogeneous when considering them in the structure context as a whole, they can be studied as being isotropic at each point. Therefore the constitutive relationship that represents the relation between stresses and correspondent strains is:

$$
\begin{bmatrix} \sigma_x \\ \sigma_y \\ \sigma_{xy} \\ \sigma_{yz} \\ \sigma_{xz} \end{bmatrix} = \begin{bmatrix} Q_{11} & Q_{12} & 0 & 0 & 0 \\ Q_{12} & Q_{22} & 0 & 0 & 0 \\ 0 & 0 & Q_{66} & 0 & 0 \\ 0 & 0 & 0 & k.Q_{44} & 0 \\ 0 & 0 & 0 & 0 & k.Q_{55} \end{bmatrix} \tag{10}
$$

where the expressions for the elastic coefficients $Q_{ij}$ are the ones provided by the literature [30] for isotropic materials, modified to account for the materials and porosities spatial distributions defined in previous sub-sections.

The shear correction factor, here represented by $k$, was estimated using a formulation similar to the one used by M. K. Singha [31]. Imposing the necessary boundary conditions associated with the problem to be studied, this yields the usual equilibrium Equation (11):

$$
[K].\{q\} = \{F\} \tag{11}
$$

This can be used next to obtain the intended numerical solution, where [K] stands for the global stiffness matrix and {q} the generalized nodal displacements vector. The vector {F} comprises the generalized forces applied to the plate.

## 3. Results and Discussion

This section consists of two main parts; the first dedicated to performance verification applications and the second wherein the set of case studies is considered focusing on the parametric influence characterization of porous nanocomposite specimens based on the ASTM tensile and three-point bending tests. Although this work does not include an experimental component, it is considered relevant to present results based on standardized testing conditions so as to facilitate the replication and comparison of results by other researchers both in the context of numerical and/or experimental studies. Thus, the standard test method for flexural properties of polymer matrix composite materials [32] which allows the determination of flexural stiffness and strength properties of polymer matrix composites, includes two different procedures, namely the three-point loading system (Figure 5), which is simulated in this work, and a four-point loading system.

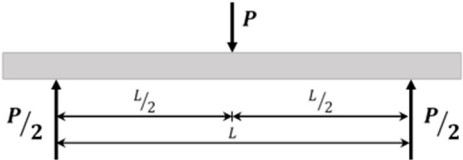

**Figure 5.** Three-Point bending test loading system.

In the case of the standard test method for tensile properties of polymer matrix composite materials [33], the aim is to determine the in-plane tensile properties of polymer matrix composite materials reinforced by high-modulus material. For this purpose, the sample should be subjected to the loading conditions illustrated in Figure 6, according to [34].

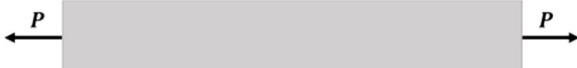

**Figure 6.** Tensile test loading system.

Thus simulating these tests with the present materials enables the specimens' mean values' responses to be predicted when considering the experimental tests.

*3.1. Verification Studies*

To guarantee the validity of the different models implemented in the studies reported in this work, some verification studies were carried out. In these studies, the results obtained were compared with the results presented by other authors. To this purpose, we considered FGM plates under a uniform transverse pressure, distributed throughout the plate upper surface.

The verification case studies consider comparisons with numerical solutions obtained by Costa and Loja [29].

The present work focused on the analysis of polymeric matrix nanocomposites for biomedical application, and allows structures to be studied of a graded introduction of MWCNTs along the thickness direction, contrary to the previous reference where the inclusion of nanotubes corresponds to a fixed percentage weight. Additionally in the present work, we considered the effects arising from the existence of porosities and concerning the shear, and the calculation of shear correction factors as a function of the materials' distribution was introduced. All these aspects; correspond to an innovative approach concerning the present model development and implementation.

3.1.1. Carbon Nanotubes

In a first case study we considered an FGM square plate with a unit length edge and an aspect ratio $a/h$ = 20. The plate, simply supported, was submitted to a uniform transverse pressure of 10 kPa. The two material phases that constitute the present FGM are respectively: one ceramic—phase A—of zirconia ($ZrO_2$) particles, and another—phase B—composed of MWCNTs dispersed in aluminum (Al) matrix. Table 1 contains the material properties of these constituents, and Table 2 presents the geometric characteristics of the CNTs.

**Table 1.** Material properties of Functionally Graded Material (FGM) constituent phases.

| Material | Properties | |
| --- | --- | --- |
| Ceramic Particles ($ZrO_2$) | $E_{(A)}$ = 200 GPa | $v_{(A)}$ = 0.3 |
| Metallic matrix (Al) | $E_{(Al)}$ = 70 GPa $v_{(Al)}$ = 0.3 | $\rho_{(Al)}$ = 2700 kg/m$^3$ |

**Table 2.** Material and geometric properties of Multi-Walled Carbon Nanotubes (MWCNT).

|  | $E$ [GPa] | $l$ [μm] | $d$ [nm] | $t$ [nm] | $\rho$ [kg/m$^3$] |
|---|---|---|---|---|---|
| **MWCNT** | 400 | 50 | 20 | 0.34 | 1350 |

The volume fraction of phase A is given by the power-law (1). The effective properties of phase B (MWCNT/Al) were estimated with Equations (5a)–(5c), (6), and (7), whereas the analogous ones for the FGM were obtained by using Equation (2). The plate was discretized in a mesh comprising ten Q4 finite elements in both in-plane directions and the maximum transverse displacements were obtained for different power-law exponents and MWCNT weight fraction. The non-dimensional transversal displacement, $\overline{w} = w/h$, and the shear correction factor obtained are presented in Table 3. Also presented is the relative deviation of the present results to the reference [29] ones, calculated according to the expression

$$\delta(\%) = \frac{w_{adim}^{present} - w_{adim}^{reference}}{w_{adim}^{reference}} * 100(\%) \tag{12}$$

where $w_{adim}^{present}$ and $w_{adim}^{reference}$ are the non-dimensional transversal displacements from the present and reference [29] studies, respectively.

**Table 3.** Results of the FGM plate for different weight fractions of MWCNT.

| $w_{MWCNT}$ [%] | Power-Law Exponent | $w_{adim}$ | $\delta$ [%] [1] | Shear Correction Factor |
|---|---|---|---|---|
| 0 | 0 (Phase A) | $3.59 \times 10^{-4}$ | — | 5/6 |
|  | 0.5 | $4.92 \times 10^{-4}$ | 0.33 | 0.8431 |
|  | 1 | $5.76 \times 10^{-4}$ | 0.36 | 0.8310 |
|  | 2 | $6.56 \times 10^{-4}$ | 0.56 | 0.7945 |
|  | 5 | $7.25 \times 10^{-4}$ | 0.44 | 0.7558 |
|  | ∞ (Phase B) | $1.03 \times 10^{-3}$ | — | 5/6 |
| 2.5 | 0 (Phase A) | $3.59 \times 10^{-4}$ | — | 5/6 |
|  | 0.5 | $4.95 \times 10^{-4}$ | 0.30 | 0.8433 |
|  | 1 | $5.84 \times 10^{-4}$ | 0.34 | 0.8309 |
|  | 2 | $6.68 \times 10^{-4}$ | 0.36 | 0.7930 |
|  | 5 | $7.42 \times 10^{-4}$ | 0.48 | 0.7521 |
|  | ∞ (Phase B) | $1.06 \times 10^{-3}$ | — | 5/6 |
| 5 | 0 (Phase A) | $3.59 \times 10^{-4}$ | — | 5/6 |
|  | 0.5 | $4.99 \times 10^{-4}$ | 0.23 | 0.8435 |
|  | 1 | $5.92 \times 10^{-4}$ | 0.61 | 0.8309 |
|  | 2 | $6.81 \times 10^{-4}$ | 0.43 | 0.7915 |
|  | 5 | $7.58 \times 10^{-4}$ | 0.50 | 0.7485 |
|  | ∞ (Phase B) | $1.10 \times 10^{-3}$ | — | 5/6 |

[1] Relative deviation to literature Reference [29].

It is relevant to note that in the present work the shear correction factors were determined for each specific plate, instead of using the constant, typical value of 5/6 that the reference considered. However, according to the results, there is a very good agreement between the results, with small relative deviations. This qualitative result in terms of deviations was expected as we considered a moderately thin plate, thus the transverse shear effects were not significant.

### 3.1.2. Porosities

For this case study, we considered a unit length square plate with an aspect ratio of $a/h = 100$, wherein zirconia particles are dispersed in an aluminum matrix according to Equation (1) with a power-law exponent of $p = 0.1$. This plate was considered fixed at the four edges and submitted to

a uniform transverse pressure. To analyze the influence of both porosity distribution and volume fraction presented in Figure 4 and Equations (3) and (4), different loads were applied to the plate. The normalized central deflection, $w_{adim}$, and load parameter, $P$, obtained by (13), were used for a better presentation of results.

$$w_{adim} = \frac{w}{h}$$
$$P = \frac{q_0 a^4}{E_m * h^4}$$

(13)

where $w$ is the central transversal displacement, $a$ represents the square length, $h$ indicates the plate thickness, and $E_m$ stand for the matrix Young's modulus. The results obtained in this case study are represented in Figure 7.

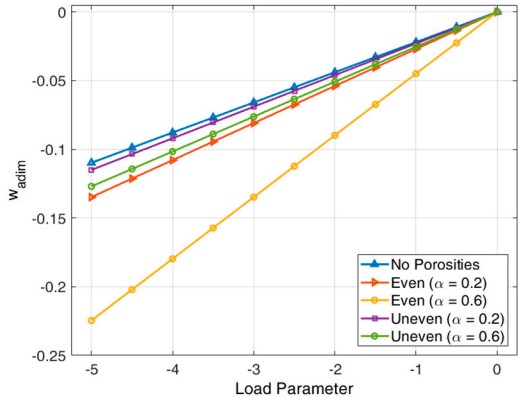

**Figure 7.** Normalized central deflection for different porosity distributions and volume fractions.

The results obtained in this study are in agreement with the literature [26], denoting higher displacements for porous FGM plates when compared with non-porous. As can be seen for the same porosity volume fraction, the even porosity distribution model promotes a larger displacement.

*3.2. Case Studies*

The present section is constituted by a set of case studies where we address the influence of the parameters that characterize the material phases and porosity distributions. Also, the influence of the MWCNTs on the resulting graded nanocomposite is assessed. In this section, all the analyses performed consider the geometrical characteristics (320/3 mm/13 mm/4mm) associated with the specimens used in the standards ASTM D 3039/D 3039M and ASTM D7264/D, corresponding respectively to a tensile test and to a three-point bending test.

3.2.1. Influence of Carbon Nanotubes Weight Fraction

The influence of the MWCNTs weight fraction on the predicted Young's modulus, estimated using Equations (2), and (5a) to (6), is depicted in Figure 8. Figure 9 aims to evidence the influence of the MWCNTs weight fraction for the same power-law exponents.

As is possible to observe from Figure 8, as the MWCNTs percent weight content increases, there is also an increasing trend in the resulting nanocomposite Young's modulus. Also visible is the effect that the exponent from the power law, that rules the material phases distribution, plays in this context. In fact when $p = 0$ this corresponds to a situation where all the plate is homogeneous and is built from a mixture of materials PSU + %MWCNT. An opposite situation occurs for $p$ = infinity where the plate is only constituted by homogeneous PSU. For all the remaining intermediate curves, we observe an evolution where the greater nanotubes addition gradient dislocates from a thickness coordinate nearer the upper surface of the plate to the lower one, when the power law exponent increases.

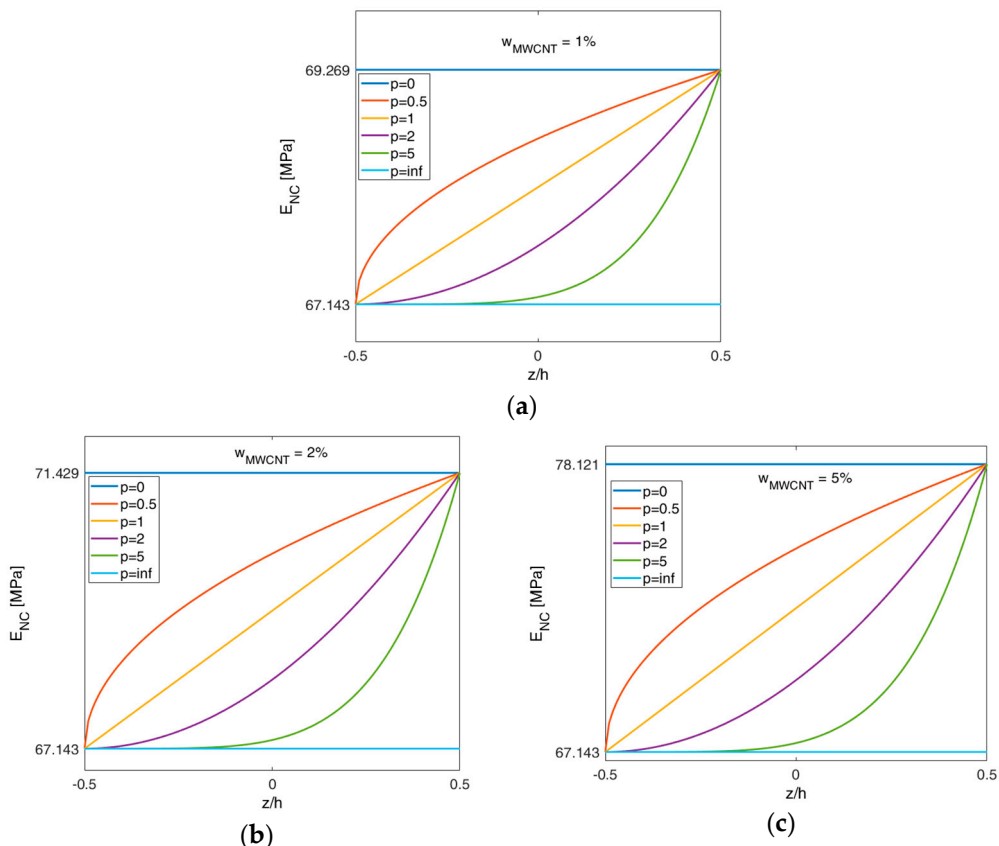

**Figure 8.** Nanocomposite Young's modulus ($E_{NC}$) evolution along the thickness direction: (**a**) $W_{MWCNT}$ = 1%; (**b**) $W_{MWCNT}$ = 2%; (**c**) $W_{MWCNT}$ = 5%.

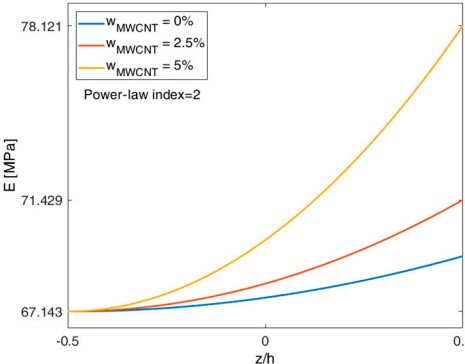

**Figure 9.** Nanocomposite Young's modulus ($E_{NC}$) evolution along the thickness direction for different MWCNTs weight fraction.

Concerning the influence of MWCNTs weight fraction, presented in Figure 9, its effect is clear in the Young's modulus of the nanocomposite constituted by PSU and nanotubes. For a specific value of the power law exponent, namely for $p$ = 2, as nanotubes are progressively mixed with PSU through the thickness, we observe that the modulus also increases, the effect being more pronounced as the weight fraction increases.

### 3.2.2. Influence of Porosity Distribution

This case study aims to evaluate the influence of volume fraction and type of distribution of porosities in an FGM PSU/(PSU + 2% MWCNT) nanocomposite, where to the coordinate $z = -h/2$ the material corresponds to PSU + 2% MWCNT and for $z = h/2$ to PSU only. For this purpose,

Figure 10 shows the power-law exponent's influence, as well as the porosity volume fraction and distribution model.

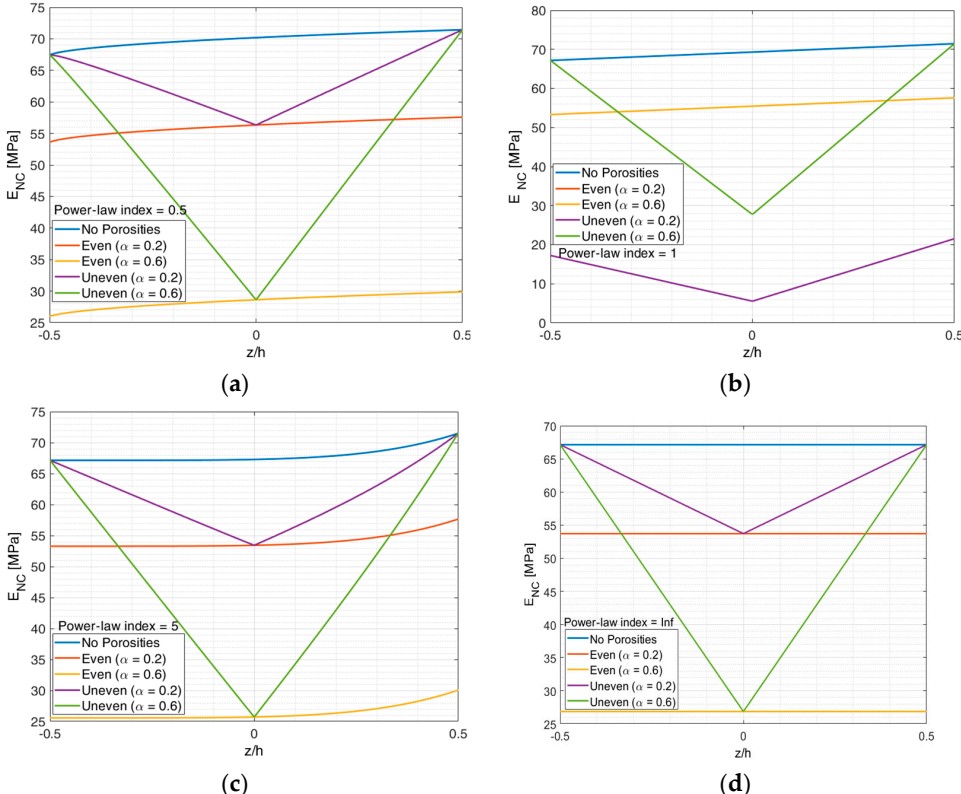

(a)                              (b)

(c)                              (d)

**Figure 10.** Nanocomposite Young's modulus ($E_{NC}$) evolution along the thickness direction for different porosity volume fractions and distributions; (**a**) p = 0.5, (**b**) p = 1, (**c**) p = 5, (**d**) p = ∞.

From this study, is evident that for the same porosity volume fraction the evolution of the considered FGM Young's modulus is very smooth for an even distribution. Due to the dependence on the absolute value of the z coordinate, the uneven porosity distribution produces a different effect on the FGM Young's modulus, making this material evolving characteristically in a different way, showing decreasing behavior on the bottom half of the thickness plate, and increasing behavior on the upper half, producing a minimal Young's modulus at the plate mid-plane and a maximum value at the outer surfaces.

3.2.3. Static Behavior of PSU + 2% MWCNTs Plate

In this section we analysed the static behavior of a nanocomposite plate, whose material results from the combination of a matrix of 15% PSU in DMF reinforced with 2% (wt) of MWCNTs. The material properties associated with these material phases are shown in Tables 4 and 5. Both studies included additional situations with and without porosity effects.

**Table 4.** Material properties of the plate's constituents.

| Material | Properties |
|----------|------------|
| Polysulfone (PSU) | $\rho$ = 1350 kg/m$^3$ |
| Dimethylformamide (DMF) | $\rho$ = 950 kg/m$^3$ |

**Table 5.** Material and geometric properties of MWCNT.

| | $E$ [GPa] | $l$ [µm] | $d$ [nm] | $t$ [nm] | $\rho$ [kg/m$^3$] |
|---|---|---|---|---|---|
| **MWCNT** | 6 | 1.4 | 70 | 5 | 2400 |

Case A: Neglecting Porosity Effect

Here, porosities were not accounted for. So, a plate made of nanocomposite PSU + 2% MWCNT without pores was considered.

- Tensile load

The plate was submitted to an axial distributed load, with 615 N/m magnitude at the free end. A maximum axial displacement of 22.901 µm was observed at the free end, as expected. Figure 11 ilustrates the axial displacement of each finite element considered for the mesh.

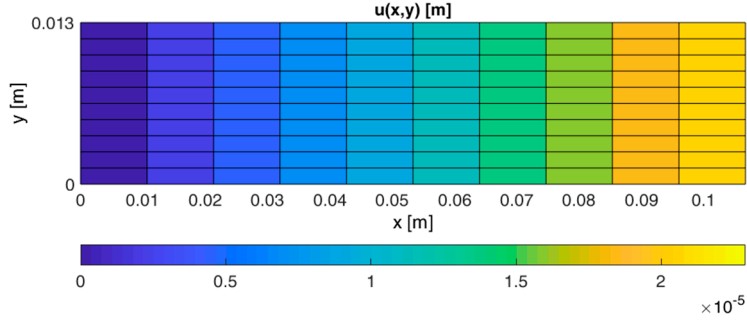

**Figure 11.** Axial displacement of a PSU + 2% MWCNT plate subjected to an axial load.

- Bending load

For this case study, the plate was subjected to a transverse line distribution load of aproximatelly 7.7 N/m. The load location and the boundary condition considered are according to the standard in Ref. [32] for a three-point loading system. The maximum transversal displacement observed was about 0.5 mm, as Figure 12 illustrates.

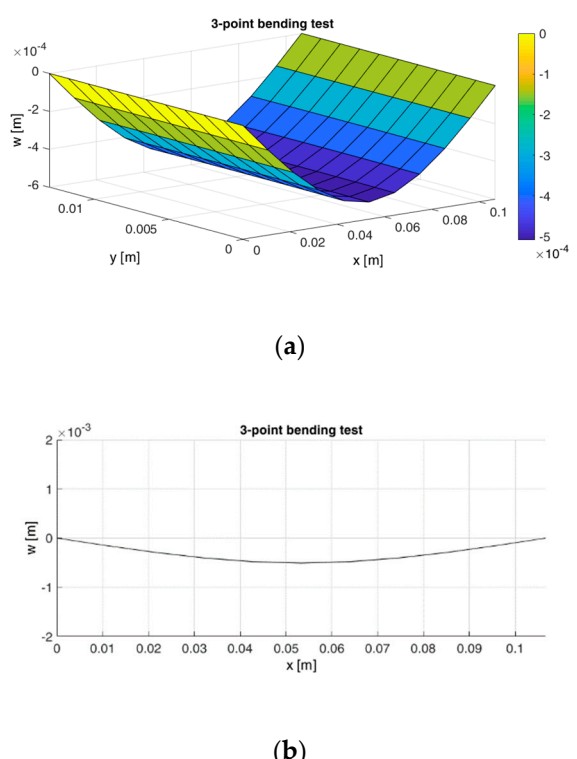

(a)

(b)

**Figure 12.** Transversal displacement of a PSU + 2% MWCNT plate: (**a**) 3D view; (**b**) view from the xz plane.

Case B: Including Porosity Effect

We consider now that porous distributions occur and may assume different types, according to Section 2.3. As previously, the tensile and three-point bending loading cases are considered, in order to characterize the behaviour of the plates in the light of this new parameter.

- Tensile load

Table 6 shows the results with respect to the study of the effect of the porosity volume fraction. To facilitate their understanding, the results were normalized using the relation $u_{adim} = u_{max}/u_{max(PSU)}$, where $u_{max(PSU)}$ is the maximum axial displacement obtained for PSU ($u_{max(PSU)} = 24.362$ μm).

**Table 6.** Normalized axial displacement for different porosity distributions.

| $\alpha$ | $u_{adim}$ | |
| --- | --- | --- |
| | **Even** | **Uneven** |
| 0 | 0.9400 | 0.9400 |
| 0.1 | 1.0451 | 0.9897 |
| 0.2 | 1.1764 | 1.0447 |
| 0.3 | 1.3451 | 1.1060 |
| 0.4 | 1.5699 | 1.1748 |
| 0.5 | 1.8846 | 1.2526 |
| 0.6 | 2.3564 | 1.3412 |
| 0.7 | 3.1425 | 1.4434 |
| 0.8 | 4.7146 | 1.5627 |
| 0.9 | 9.4300 | 1.7037 |

These results confirm the influence of the type of porosity distribution already described in one of the verification studies. The axial displacement for an uneven porosity distribution presents a notorious smoother variation with the porosity volume fraction. To note additionally is that as expected, for the null value of $\alpha$, the normalized axial displacement is lower than unity as the PSU + 2% MWCNT plate has a higher axial stiffness when compared to the PSU plate.

- Bending load

The maximum transversal displacements observed for different porosity volume fraction and distribution are presented now in Table 7. As in the previous case, the results were normalized by $w_{adim} = w_{max}/w_{max(PSU)}$, where $w_{max(PSU)}$ represents the maximum transversal displacement obtained for a PSU plate under the same conditions ($w_{max(PSU)} = 541.22$ μm).

**Table 7.** Normalized transversal displacement for different porosity distributions.

| $\alpha$ | $w_{adim}$ | |
| --- | --- | --- |
| | **Even** | **Uneven** |
| 0 | 0.9400 | 0.9400 |
| 0.1 | 1.0443 | 0.9642 |
| 0.2 | 1.1746 | 0.9896 |
| 0.3 | 1.3421 | 1.0163 |
| 0.4 | 1.5654 | 1.0448 |
| 0.5 | 1.8779 | 1.0745 |
| 0.6 | 2.3466 | 1.1062 |
| 0.7 | 3.1277 | 1.1400 |
| 0.8 | 4.6896 | 1.1762 |
| 0.9 | 9.3751 | 1.2160 |

Also in the bending case, the results indicate the same evolution for the porosity volume fractions and distributions considered. A similar remark concerning the normalized transverse displacement for the null value of $\alpha$, can be here expressed.

### 3.2.4. Static Behavior of FGM Plate—PSU/(PSU + 2% MWCNTs)

To evaluate the influence of MWCNTs distribution along the plate thickness, this case study considers a graded mixture, where the matrix (PSU)—Phase B—is reinforced with MWCNTs, whose volume fraction varies according to Equation (1). In this case a misture of PSU + 2% MWCNT is considered the phase A of the FGM. This plate is also considered under the same loading situations and boundary conditions as the ones applied to the PSU + 2% MWCNT plate previously studied.

Case C: Neglecting Porosity Effect

In this first point, the plate is considered free of porosities, and the studies under tensile and bending loads are presented for different loads and power-law exponents.

- Tensile load

The results obtained in this study are presented in a normalized form using the same expression as in Case A, presented in Table 8. These results allow the conclusion that as the power-law exponent increases, the normalized axial displacement also increases, achieving a unit value for the case where the material coincides with PSU, phase B.

**Table 8.** Normalized axial displacement for different power-law exponents.

| Power-Law Exponent | $u_{adim}$ |
|:---:|:---:|
| 0 (A) | 0.9400 |
| 0.5 | 0.9592 |
| 1 | 0.9691 |
| 2 | 0.9792 |
| 5 | 0.9895 |
| $\infty$ (B) | 1.0000 |

- Bending load

In this case, the normalized results for the transversal maximum displacement are presented in Table 9.

**Table 9.** Normalized transversal displacement for different power-law exponents.

| Power-Law Exponent | $w_{adim}$ |
|:---:|:---:|
| 0 (A) | 0.9400 |
| 0.5 | 0.9616 |
| 1 | 0.9694 |
| 2 | 0.9754 |
| 5 | 0.9823 |
| $\infty$ (B) | 1.0000 |

A similar interpretation is here applicable when considering the normalized transverse displacement evolution.

Case D: Including Porosity Effect

The nanocomposite FGM plate is now considered to contain pores, thus their effects are now considered in the present case.

- Tensile load

For this case, the normalized results obtained for even and uneven porosity distributions are the ones presented in Tables 10 and 11.

**Table 10.** Normalized maximum axial displacement considering even porosity distribution.

| | $u_{adim}$ | | | | | |
|---|---|---|---|---|---|---|
| | **Power-Law Exponent (p)** | | | | | |
| $\alpha$ | **0 (A)** | **0.5** | **1** | **2** | **5** | **∞ (B)** |
| 0 | 0.9400 | 0.9592 | 0.9691 | 0.9792 | 0.9895 | 1.0000 |
| 0.1 | 1.0451 | 1.0652 | 1.0774 | 1.0899 | 1.1027 | 1.1158 |
| 0.2 | 1.1764 | 1.1974 | 1.2128 | 1.2286 | 1.2448 | 1.2616 |
| 0.3 | 1.3451 | 1.3666 | 1.3867 | 1.4074 | 1.4288 | 1.4508 |
| 0.4 | 1.5699 | 1.5911 | 1.6185 | 1.6468 | 1.6761 | 1.7065 |
| 0.5 | 1.8846 | 1.9036 | 1.9429 | 1.9838 | 2.0264 | 2.0710 |
| 0.6 | 2.3564 | 2.3682 | 2.4293 | 2.4936 | 2.5613 | 2.6329 |
| 0.7 | 3.1425 | 3.1321 | 3.2398 | 3.3551 | 3.4789 | 3.6122 |
| 0.8 | 4.7146 | 4.6220 | 4.8604 | 5.1244 | 5.4191 | 5.7496 |
| 0.9 | 9.4300 | 8.8129 | 10.8300 | 10.8394 | 12.2469 | 14.0751 |

**Table 11.** Normalized maximum axial displacement for uneven porosity distribution.

| | $u_{adim}$ | | | | | |
|---|---|---|---|---|---|---|
| | **Power-Law Exponent (p)** | | | | | |
| $\alpha$ | **0 (A)** | **0.5** | **1** | **2** | **5** | **∞ (B)** |
| 0 | 0.9400 | 0.9592 | 0.9691 | 0.9792 | 0.9895 | 1.0000 |
| 0.1 | 0.9897 | 1.0094 | 1.0203 | 1.0315 | 1.0430 | 1.0529 |
| 0.2 | 1.0447 | 1.0649 | 1.0771 | 1.0895 | 1.1023 | 1.1114 |
| 0.3 | 1.1060 | 1.1266 | 1.1403 | 1.1543 | 1.1686 | 1.1766 |
| 0.4 | 1.1748 | 1.1957 | 1.2111 | 1.2270 | 1.2432 | 1.2498 |
| 0.5 | 1.2526 | 1.2738 | 1.2913 | 1.3093 | 1.3278 | 1.3325 |
| 0.6 | 1.3412 | 1.3626 | 1.3827 | 1.4034 | 1.4247 | 1.4268 |
| 0.7 | 1.4434 | 1.4648 | 1.4881 | 1.5121 | 1.5369 | 1.5356 |
| 0.8 | 1.5627 | 1.5837 | 1.6110 | 1.6392 | 1.6684 | 1.6624 |
| 0.9 | 1.7037 | 1.7240 | 1.7564 | 1.7901 | 1.8251 | 1.8125 |

- Bending load

Tables 12 and 13 reports the results obtained for the maximum transversal displacement in this study.

**Table 12.** Normalized maximum transversal displacement for even porosity distribution.

| | $w_{adim}$ | | | | | |
|---|---|---|---|---|---|---|
| | **Power-Law Exponent (p)** | | | | | |
| $\alpha$ | **0 (A)** | **0.5** | **1** | **2** | **5** | **∞ (B)** |
| 0 | 0.9400 | 0.9616 | 0.9694 | 0.9754 | 0.9823 | 1.0000 |
| 0.1 | 1.0443 | 1.0674 | 1.0770 | 1.0845 | 1.0930 | 1.1109 |
| 0.2 | 1.1746 | 1.1994 | 1.2115 | 1.2210 | 1.2317 | 1.2496 |
| 0.3 | 1.3421 | 1.3686 | 1.3845 | 1.3969 | 1.4109 | 1.4278 |
| 0.4 | 1.5654 | 1.5935 | 1.6152 | 1.6322 | 1.6512 | 1.6653 |
| 0.5 | 1.8779 | 1.9071 | 1.9385 | 1.9629 | 1.9903 | 1.9978 |
| 0.6 | 2.3466 | 2.3748 | 2.4240 | 2.4624 | 2.5052 | 2.4964 |
| 0.7 | 3.1277 | 3.1477 | 3.2358 | 3.3049 | 3.3808 | 3.3273 |
| 0.8 | 4.6896 | 4.6712 | 4.8734 | 5.0331 | 5.2042 | 4.9890 |
| 0.9 | 9.3751 | 9.1076 | 9.9824 | 10.7094 | 11.4165 | 9.9736 |

**Table 13.** Normalized maximum transversal displacement for uneven porosity distribution.

| | $w_{adim}$ | | | | | |
|---|---|---|---|---|---|---|
| | **Power-Law Exponent (p)** | | | | | |
| $\alpha$ | **0 (A)** | **0.5** | **1** | **2** | **5** | **∞ (B)** |
| 0 | 0.9400 | 0.9616 | 0.9694 | 0.9754 | 0.9823 | 1.0000 |
| 0.1 | 0.9642 | 0.9862 | 0.9943 | 1.0007 | 1.0079 | 1.0257 |
| 0.2 | 0.9896 | 1.0119 | 1.0206 | 1.0273 | 1.0349 | 1.0526 |
| 0.3 | 1.0163 | 1.0391 | 1.0482 | 1.0553 | 1.0633 | 1.0812 |
| 0.4 | 1.0448 | 1.0677 | 1.0774 | 1.0849 | 1.0934 | 1.1116 |
| 0.5 | 1.0745 | 1.0980 | 1.1082 | 1.1162 | 1.1252 | 1.1429 |
| 0.6 | 1.1062 | 1.1301 | 1.1410 | 1.1495 | 1.1590 | 1.1768 |
| 0.7 | 1.1391 | 1.1641 | 1.1762 | 1.1849 | 1.1943 | 1.2118 |
| 0.8 | 1.1792 | 1.2010 | 1.2134 | 1.2232 | 1.2341 | 1.2548 |
| 0.9 | 1.2160 | 1.2409 | 1.2545 | 1.2654 | 1.2774 | 1.2936 |

When considering simultaneously porosities and power-law effects in the response of the plate, both for tensile and bending studies, the results demonstrate that the displacements increase with the increase of both parameters. However, for lower values of the considered parameters, the plate presented displacements lower or slightly above the one obtained when considering a PSU plate (without MWCNTS).

## 4. Conclusions

The present work aimed to characterize the effect that the inclusion of nanoparticles would have in a polymeric material used in biomedical applications. This assessment was performed by considering loading cases corresponding to two ASTM common tests, and it was possible to conclude that by integrating MWCNTs the plate specimens become stiffer which yielded lower values of displacements when they were submitted to tensile or bending loads. This trend has already been observed when carbon nanotubes are added to other types of materials, where it was also possible to conclude that within the range of weight fractions considered, the higher the corresponding volume fraction, the higher the Young's modulus of the resulting nanocomposite would become. This trend is accompanied by the properties continuous variation dictated by the mixture distribution law used, which was the exponent power law.

Concerning the pore distributions and their influence on the FGM Young's modulus it was possible to conclude that from the distributions analyzed that the uneven porosity distribution produced an effect on the material property, which led to a minimal Young's modulus at the plate mid-plane and a maximum value at the outer surfaces, due to the dependence on the absolute value of the z coordinate. On the other hand, the pores even distribution, produced a very smooth evolution of the FGM Young's modulus.

These two different parameters were analyzed in a separate way and then in a joint manner, for the same loading conditions. From the results obtained it is possible to understand that as the porosity level increases when the pores distribution is uneven, the axial and transversal displacements reflect also a smooth variation with the porosity volume fraction, in contrast to the even distribution.

From the studies developed and as an overall conclusion; the carbon nanotubes and the porosity distributions analyzed, provide different and opposite effects in the context of the nanocomposite materials analyzed.

**Author Contributions:** All authors contributed to the design and implementation of the research, to the analysis of the results, and to the writing of the manuscript.

**Funding:** This research was funded by Project IPL/2019/MOCHVar/ISEL.

**Acknowledgments:** This work was supported by FCT, through IDMEC, under LAETA, project UID/EMS/50022/2019. The authors also acknowledge the support of Project IPL/2019/MOCHVar/ISEL.

**Conflicts of Interest:** The authors declare no conflict of interest.

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
