# Peer review of "Mechanical Behavior of Porous Functionally Graded Nanocomposite Materials"

_carbon, 2019_

Round 1

Reviewer 1 Report

In this paper, the mechanical behaviour of porous functionally graded nanocomposite materials is investigated. Numerical finite element simulations based on the first order shear deformation theory are performed for the analysis of plate-type structures. A parametric analysis is carried out in order to analyse the influence of FGM, porosities and MWCNTs distributions and weight fractions on the Young’s modulus value and static behaviour of the plate.

This paper is well written and rightly organised, the mathematical models are clearly described, the numerical results of the parametric analysis are properly described by means of figures. Moreover, the English Language throughout the text is very good.

It must be underlined that the different models implemented for the FGM, porosities and MWCNTs distributions are properly verified by means of comparisons with data retrieved from the pertinent literature.

In addition, it is interesting the observation related to the opposite effects of carbon nanotubes and porosities distributions on the static behaviour of the plate.

In order to consider this paper for publication, there is only one relevant aspect that should be carefully taken into account by the Author.

In the Introduction, in addition to Ref. [2] (functionally graded coatings for thermal insulation), it must be underlined also the importance of the functionally graded coatings for vibration reduction: for example, multi-layered functionally graded composites can be used in order to get mechanical components with reduced vibrations by increasing the damping behaviour at the layer interfaces.

To this aim, the following relevant papers should be added and discussed by the Authors:

a) Catania, G.; Strozzi, M. Damping oriented design of thin-walled mechanical components by means of multi-layer coating technology. Coatings 2018, 8, 73.

b) Yu, L.; Ma, Y.; Zhou, C.; Xu, H. Damping efficiency of the coating structure. International Journal of Solids and Structures 2005, 42, 3045-3058.

Therefore, in the opinion of the Reviewer, by considering the previous notes, the paper should be accepted for publication after minor revision.

Author Response

Dear Reviewer

We are very grateful to the reviewer for the comments, which were very important for us to understand that some aspects would need to be better detailed and/clarified.

We have addressed all the comments as it will be possible to see in the attached file, and we have highlighted them in yellow in the revised manuscript.

We renew our sincere thanks for the comments we received, hoping that the actions we took to respond, meet the expectations.

Reviewer 2 Report

Review of

Mechanical behavior of porous functionally graded nanocomposite materials”

by A. F. Mota and M. A. R. Loja

Contents

In the manuscript, axial and flexural behaviors of porous functionally graded (FG) plates are investigated by first-order shear deformation theory. The finite element method is exploited to examine the structural responses of FG plates. A numerical study is accomplished to analyze the effects of material parameters on structural responses.

Comments

The subject is of interest for the journal readership.

A revision is required to tackle the following points.

·    Novelties and merits of the present study should be clearly illustrated in comparison with a previous paper of the same authors [22]:

“D. M. S. Costa and M. A. R. Loja, “Assessing the static behavior of hybrid CNT-metal-ceramic composite 551 plates,” AIMS Mater. Sci., vol. 3, no. 3, pp. 808–831, 2016”.

The authors in the aforementioned research presented a detailed analysis of the effects of the material parameters on the transverse displacement and stress distributions of the functionally graded first-order shear deformable plate. Employing the same solution technique and structural formulation, in the presented study FG plates with different porosities distributions are investigated.

However, the structural response of FG plates integrating porosities distributions have been extensively examined in literature. Comparisons should be performed also with the contributions already cited in the submitted manuscript, see  Kim et al. (2019) as Ref. [18], Nguyen et al. (2018) as Ref. [19], Merdaci & Belghoul (2019) as Ref. [20].

According to the authors (at line 232), the main motivation of present study is to examine structural behaviors of FG porous plates in the specific context of materials used for biomedical applications. However, no indication seems to be made in the submitted paper that would suggest that the present work is in any way different from that which have preceded it in terms of significance.

·    ASTM standard tests, such as simple tension test (ASTM D3039) and three-point bending test (ASTM D 7264), can be efficiently used to perform experimental measurements. These standard tests are not introduced with the aim of application in the analytical or numerical analysis.

·    While it is emphasized several times in the manuscript that the structural response of plates made of porous functionally graded nano-composite materials will be examined but the isotropic material behavior is lastly assumed in the analysis, as explicitly mentioned in line 271.

·    The presented results for the axial deformation of the plate as illustrated in Figure 9 can be conveniently detected employing the simple rod model. However, it should be noted that in FG plates, there exists a flexural-stretching coupling that certainly affects the axial response of the plate. Moreover, the presented results for the transverse deformation of the plate as illustrated in Figure 10 can be conveniently detected employing the simple beam model. However, the FG plates not only exhibit the flexural-stretching coupling effects but also demonstrate the transverse and twisting curvatures.

·    The introductory section should be improved by providing a more significant overview of the state of the art on mechanical behaviour of nanocomposites. In particular, recent contributions on stress-driven nonlocal integral modeling of elastic nanobeams and nanoplates should be consulted, referred to and added to the list of references.

With these improvements, the manuscript can be reconsidered for publication in Carbon.

Author Response

(The authors gave the same response as above.)

Round 2

Reviewer 2 Report

The paper is now suitable for publication in Carbon.